# Community engagement in mass drug administration participatory interventions: A scoping review

Anouk Chouaïd[1]*, Sarah Louart[2], Adama Faye[3], El Hadj Ba[4], Jordi Landier[5], Valéry Ridde[1,3]

1 Université Paris Cité and Université Sorbonne Paris Nord, IRD, Inserm, Ceped, Paris, France, 2 Univ. Lille, CNRS, UMR - CLERSE - Centre Lillois d'Etudes Et de Recherches Sociologiques Et Economiques, Lille, France, 3 Institute of Health and Development, Cheikh Anta Diop University, Fann-Dakar, Senegal, 4 Institut de Recherche pour le Développement, Dakar, Sénégal, 5 Aix Marseille Univ, Institut de Recherche pour le Développement, INSERM, SESSTIM, ISSPAM, Marseille, France

* anoukchouaid@gmail.com

## Abstract

Community engagement (CE) has been increasingly acknowledged as a pivotal element in the efficacy of mass drug administration (MDA) programs designed to control and eradicate diseases such as malaria, lymphatic filariasis, and onchocerciasis. The objective of this scoping review was to evaluate the contribution of community engagement (CE) to participatory interventions for mass drug administration (MDA). A systematic search of six databases yielded 32 articles, spanning 24 MDA interventions across 20 countries, primarily in low- and middle-income settings. The review indicates that most CE initiatives are constrained to passive information dissemination, with a paucity of initiatives that prioritize active community involvement or decision-making. The review identified several key challenges, including a lack of clear definitions for CE, inconsistent objectives, and difficulties in evaluating its effectiveness. Despite evidence that CE can enhance MDA coverage and compliance, the lack of standardized frameworks hinders comprehensive evaluation and comparison across studies. Furthermore, only a minority of interventions involved communities in the design or evaluation stages of MDA programs. The findings emphasize the importance of context-specific approaches, especially in addressing local sociocultural dynamics and including marginalized populations. Future interventions should prioritize sustainable capacity-building and adopt participatory frameworks that promote shared decision-making. Addressing these challenges could enhance the effectiveness of MDA campaigns and improve health outcomes in affected communities.

**Data availability statement:** All data underlying the findings described in the manuscript is available either within the manuscript itself or in the documents provided as supplementary information. Protocols also available at DOI: dx.doi.org/10.17504/protocols.io.8epv5xq15g1b/v1.

**Funding:** Financial support from the French Agence Nationale de Recherche (grant no. ANR-23-CE35-0002-01) and from L'Initiative (grant no. 23SANIC216), implemented by Expertise France, was limited to funding a single field mission aimed at gaining an understanding of the study context.

**Competing interests:** The authors have declared that no competing interests exists.

## Author summary

- The persistence of a public health approach that exploits community approaches is confirmed in the MDA field. The lack of clear definitions and standardized approaches limits the ability to assess and compare CE interventions.

- This scoping review highlights the difficulties in implementing community engagement (CE) in mass drug administration (MDA) studies, with most interventions focusing on (passive) information dissemination rather than genuine (active) community involvement.

- Understanding local sociocultural contexts and ensuring the inclusion of marginalized groups are critical for enhancing participation and equity in MDA programs.

- Future interventions should prioritize sustainable, capacity-building approaches to maximize the effectiveness of CE in MDA efforts.

## Introduction

The fight against vector-borne diseases remains a major global health concern, particularly in low- and middle-income countries (LMICs), where malaria and neglected tropical diseases (NTDs) still cause significant morbidity and mortality [1,2]. The World Health Organization (WHO) has made the control, elimination, and eradication of these diseases a priority through its NTD Roadmap 2021–2030, and the Global Technical Strategy for Malaria 2016–2030 [3–5]. A range of public health strategies are used to fight these diseases, including screening and treatment, preventive chemotherapy, and mass drug administration (MDA) [4].

MDA is defined as the administration of drugs to an entire population (or subpopulation) in a given area, regardless of individual disease status, to reduce or interrupt transmission [6]. MDA is a well-established intervention in the field of NTDs, implemented at scale for decades to control or eliminate diseases such as lymphatic filariasis, onchocerciasis, and trachoma [6]. These interventions are often nationally coordinated and rely on community-based distribution [7,8]. In contrast, MDA for malaria is less common and has mostly been implemented in research settings [9]. This is partly because significant progress has been made in the fight against malaria through vector control, but also through diagnosis and treatment of symptomatic individuals, as well as through preventive strategies such as seasonal malaria chemoprevention (SMC) [10]. However, in near-elimination contexts, advances in molecular diagnostics have revealed a large proportion of asymptomatic carriers harboring low-density Plasmodium infections. In the absence of highly sensitive point-of-care diagnostics, MDA remains the only feasible strategy for targeting these hidden reservoirs and accelerating progress toward malaria elimination [11,12]. In West Africa, and particularly in Senegal, interest in MDA has increased following the successful implementation of SMC in children under five [13]. This seasonal intervention, based on the monthly administration of antimalarials during the transmission season, was introduced in the early 2010s and has since been scaled up in over 13 countries targeting 50 million children [3,14]. Building on these experiences, research studies are now exploring the efficacy of

MDA as a complementary tool in the fight against malaria. In this context, learning from past experiences of MDA to inform future interventions is crucial for ensuring ethical and effective action. Community engagement has been less documented in malaria MDA compared to other diseases, which makes it a relatively new and emerging practice in this field; drawing lessons from previous MDA experiences is therefore essential.

One of the recurring lessons from past MDA interventions is the pivotal role of community engagement (CE) in achieving both high coverage and local acceptability while also ensuring ethically sound research practices.

High participation levels are necessary for MDA to achieve an impact at the population level [15]. In these cases, community trust, understanding, and support become even more critical [16]. Moreover, CE is increasingly recognized as a crucial element of health research due to its contribution to more ethical, relevant, and well-conducted research. Community-based participatory (CBP) intervention theory suggests that engaging community members as collaborators contributes to the relevance, acceptability, and effectiveness of health interventions [17,18]. More broadly, CE is defined as "a process of working collaboratively with groups of people linked by geography, interest or health issues to address social and health challenges affecting them" [19]. In the context of MDA, CE refers to the involvement of communities at all stages of intervention - design, planning, implementation, and evaluation [20].

CE strategies have been particularly emphasized in NTD programs, where they are often embedded in long-standing MDA campaigns. In recent years, a growing body of literature has also emerged describing CE in malaria MDA studies, particularly in the context of research or elimination settings. However, the goals, implementation, and evaluation of CE remain inconsistently described across studies. For example, the distinction between ethical and instrumental goals of engagement is often unclear, and the impact of CE on outcomes is rarely systematically assessed [21].

A previous systematic review published in 2016 described community involvement in malaria MDA studies published up to 2013 and called for more structured evaluations of engagement practices [22]. To our knowledge, no comprehensive review has been conducted since then. Moreover, the number of published MDA studies increased significantly over the past decade, particularly those describing participatory approaches in both NTD and malaria contexts. At the same time, the concepts and practices of CE have evolved, raising new questions about its goals, modalities, and impact.

In light of these developments, we conducted a scoping review to identify how community engagement is mobilized in participatory interventions implemented in the context of MDA. While we acknowledge the differences in implementation between malaria and NTDs, this review seeks to highlight common practices, contextual adaptations, and gaps in reporting and evaluation. By doing so, we aim to contribute to a better understanding of CE strategies and inform future MDA efforts across disease contexts.

## Methods

We conducted a scoping review on CE in MDA interventions following Arksey and O'Malley [23] method and we followed each distinct stage of the framework of the WHO guide [24]. We chose this type of review because it allowed us to: (i) have an overview of the available evidence in a given field, (ii) identify key characteristics or factors related to a concept and (iii) identify and analyze gaps in the knowledge base [25]. The reporting was guided by the preferred reporting items for systematic reviews and meta-analyses (PRISMA) checklist [26], as described in Fig 1. We registered the protocol in a dedicated platform (protocols.io, DOI: dx.doi.org/10.17504/protocols.io.8epv5xq15g1b/v1 (Private link for reviewers: https://www.protocols.io/private/2391AFC090A411EFBAC80A58A9FEAC02 to be removed before publication.)). The protocol and the full search strategy are available in the supplementary materials (S1 Text, S2 Text).

## Research strategies

The following research question guided our review: "What is known about community engagement in participatory interventions for mass drug administrations?". We listed 2 keywords related to the key concepts of our research question: mass drug administration (MDA) and community engagement (CE).

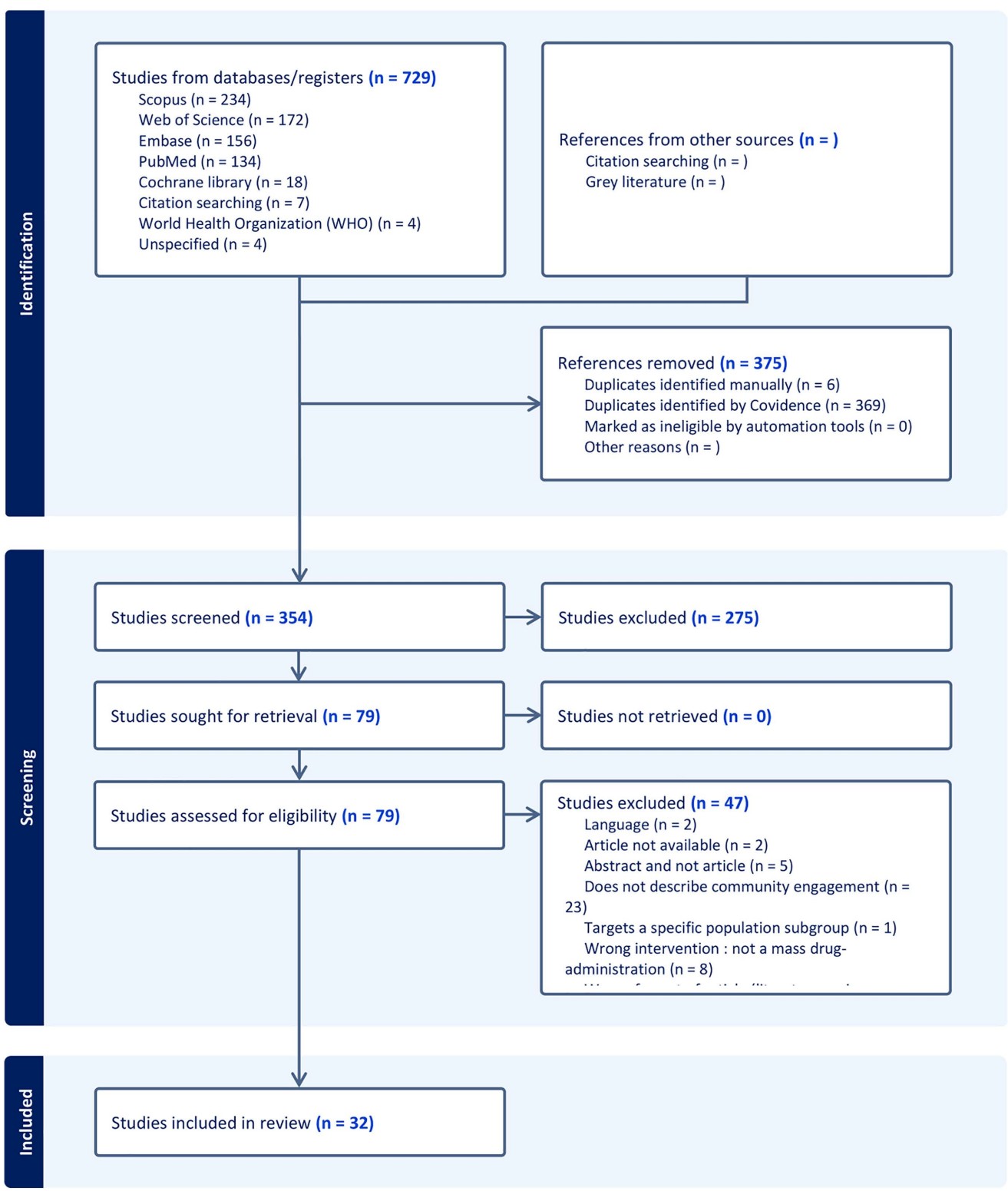

**Fig 1. PRISMA (Preferred Reporting Items for Systematic Reviews and Meta-Analyses) Checklist.**

We defined MDA as the administration of a treatment, irrespective of the knowledge of symptoms or presence of infection, to an entire population in a given area [27]. To ensure that the interventions were homogeneous and involved the general and asymptomatic population, we excluded interventions implying the administration of drugs only to infected people such as (i) interventions of mass screening and treatment (MSAT) and focal screening and treatment (FSAT), that require testing all people in a geographical area and treating only positive cases, (ii) interventions of focal MDA (also called focal drug administration or targeted MDA), which refers to reactive interventions triggered by a clinical case and propose treatment to asymptomatic persons around the infected one, and (iii) interventions targeting a specific population subgroup, such as seasonal malaria chemoprevention [28,29].

We screened all articles involving interventions described as participatory, community-based or involving CE. Then, abstracts and full texts were reviewed, and we (AC, VR) selected articles about initiatives of community engagement (see Appendix for definitions and full search strategy). Based on the assumption that there is a paucity of scientific literature on this [22], we (AC, SL, VR) decided not to limit the review to references that evaluated the effectiveness of the interventions.

We identified relevant literature in January 2024 by searching for articles based on combinations of our keywords in six scientific databases (PubMed, Embase, Cochrane library, Global Index Medicus, Scopus, and Web of Science). We adapted the search strategy, including each database's identified keywords and index terms. We collected and imported all identified citations into Covidence, where we removed duplicates.

We (AC and VR) conducted a first screening based on the titles and abstracts, followed by a full-text selection. We discussed all uncertainties collectively. To be included, the references had to meet the following criteria: (i) the article was a program or project report or study that implemented CE during a MDA intervention, (ii) the intervention was described by its authors as participatory or became so during the implementation process, (iii) the article was empirical and peer-reviewed, (iv) the text was written in French or English, (v) the full text was available.

Following the text selection from the databases, we searched for any additional articles in the reference list of included studies. Using Covidence, we extracted data on the characteristics of articles (authors, year of publication, etc.), the characteristics of MDA interventions, of the CE interventions, and, if present, the evaluation of the intervention (type of evaluation, method, results, etc.).

For data extraction and analysis, we (AC, VR) relied on two theoretical frameworks checklist: the MMAT (Mixed Methods Appraisal Tool) to describe the methodological characteristics of the studies, and the TIDieR (Template for intervention description and replication) to describe and report the key features of the interventions. Full details are available in the supplementary material (S1 Data, S2 Data).

## Patients and public involvement statement

Patients and/or the public were not involved in the design, or conduct, or reporting, or dissemination plans of this research.

## Results

### Literature search and study selection

Our search strategy yielded 729 citations overall. We excluded 375 duplicate records, leaving 354 articles. Based on their titles and abstracts, we selected 79 of these to undergo full-text screening. A final set of 32 articles met all inclusion criteria and were selected for analysis (Fig 1). The main reason for excluding articles was that they did not report any CE.

The included articles were published between 1986 and 2023, with half published after 2017 (Figs 2 and 3). They reported 24 CE interventions in 20 countries: American Samoa, Benin, Cambodia, Cameroon, Fiji, Haiti, India, Indonesia, Kenya, Laos, Mali, Myanmar, Nicaragua, Nigeria, Sierra Leone, Tanzania, Uganda, Vanuatu, Vietnam, Zambia. The

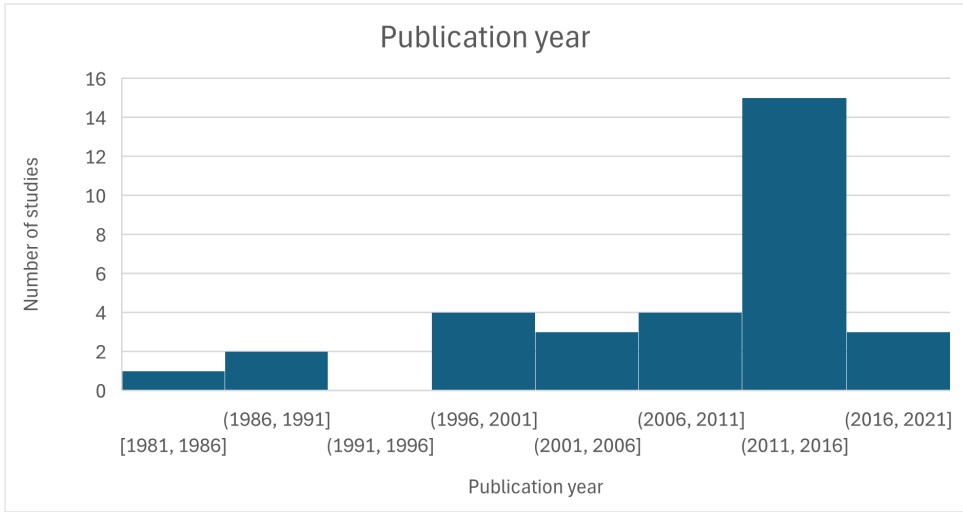

**Fig 2. Publication year.**

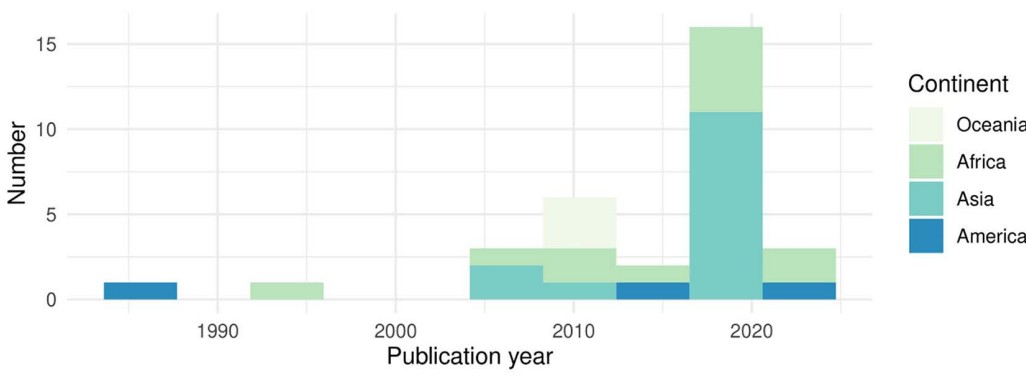

**Fig 3. Geographical and time distribution of the studies included.**

geographic distribution of the studies included is shown in Fig 3. Most took place in Asia (14/32, 44%) or Africa (12/32, 37%). The intervention occurred in a low-income country (LIC) for 13 out of 20 countries (65%), LMIC for 5/20 (25%) and UMIC for 2/20 (10%) -defined by the World Bank for the current 2022 fiscal year. Several articles were published on the same intervention.

## MDA interventions

Of the 24 interventions, 16 focused on initiatives to tackle neglected tropical diseases (NTDs): 9 targeted lymphatic filariasis [28–36], 3 onchocerciasis [37–39] and 4 multiple infections including lymphatic filariasis, onchocerciasis, soil-transmitted helminths (STH), schistosomiasis, and trachoma [40–43]. The other 8 initiatives targeted malaria [44–58]. Among those 8 interventions, 4 belonged to the same research project, the targeted malaria elimination (TME), a pilot program that included MDA interventions across the Greater Mekong sub-Region, in Cambodia, Laos, Myanmar, and Vietnam. These 4 interventions were at the source of 11 articles: 3 on the intervention in Cambodia [44,46,47], 3 in Laos [49–51], 3 in Myanmar [52–54], 1 in Vietnam [56]. One article compared implementation in the 4 countries [45].

Six interventions were conducted at a national level, 14 at a level of one or more districts, 4 interventions involved less than 5 villages. They took place between 1981 and 2020, with half of them after 2013. The interventions were implemented in rural (n = 15), urban (n = 2) and both rural and urban (n = 7) settings. Most interventions excluded pregnant women, sick people, and small children under the age of two, but the rest of the population was targeted. Interventions for onchocerciasis consisted in the administration of ivermectin, those for lymphatic filariasis of diethylcarbamazine alone (n = 1), or diethylcarbamazine combined with albendazole (n = 3) or ivermectin (n = 3). Interventions involving multiple NTDs involved the administration of praziquantel, albendazole, ivermectin, and azithromycin (n = 1). Among mass administrations of antimalarial drugs, two studies were older. The first reported on a 1981 intervention involving the administration of chloroquine + primaquine for 3 days in Nicaragua [55]; the second was carried out in Vanuatu in 1991, with the weekly administration of primaquine, chloroquine, pyrimethamine and sulfadoxine for 9 weeks [57]. Other malaria interventions were more recent and involved the administration of dihydroartemisinin/piperaquine [58], with the addition of primaquine in interventions carried out in the Mekong [44–47,49–54,56]. At all TME sites, MDA followed a similar design: intervention villages received MDA of dihydroartemisinin piperaquine for three days and a single low dose primaquine every month for three months. A Haitian study did not specify the drugs, as the MDA intervention was cancelled due to several cases of severe side effects after MDA in a neighboring commune as well as the COVID-19 pandemic [48].

Governments funded most mass administration interventions (n = 10), followed by NGOs or US or UK government agencies (n = 8), or by joint government/NGO funding (n = 3). The United Nations financed one intervention, and two studies did not specify the source of funds for the intervention.

## Notion of community engagement

Of the 32 articles, 18 explicitly mentioned "community engagement". Other terms used were "community participation" (n = 7), "community-directed interventions" (n = 3), "social mobilization" (n = 2), "community mobilization" (n = 2), "community involvement" (n = 1), "community empowerment" (n = 1), "active social participation" (n = 1), "stakeholder engagement" (n = 1). All articles mentioning the term "community engagement" were published between 2017 and 2023, and 13 out of 18 addressed antimalarial MDAs (including 11 articles on the TME in Mekong).

Only a few authors (n = 6) defined CE, and they all reported on the TME projects. The definitions emphasized the involvement of local stakeholders. Bardosh et al. defined CE as "the process of engaging those who are affected by a particular problem in the process of solving and mitigating that problem" [48]; Adhikari et al. as the "process of working collaboratively with relevant partners who share common goals and interests" [50]. In contrast, some authors offered a more concrete description, as Pell et al., who defined CE as « a range of activities, such as employing local people, including community health workers and other field workers, seeking support from village leaders and offering health education » [45]. Adhikari et al., proposed to refer to a theoretical framework, where CE is divided in five key elements: "1) stakeholder and authority engagement; 2) enlisting local human resources; 3) utilizing formative research prior to conducting MDA trials; 4) responsiveness and adapting to local challenges as they arise; and 5) sharing control and leadership with the community in deciding and organizing activities" [50].

The most common justification stated for CE interventions was the improvement of population coverage (in 24 out of the 26 studies in which a rationale was provided). Intermediate objectives aiming to increase participation were also cited: increasing "acceptability", "compliance", "gain community's trust". Other principles, such as "sustainability", "equity" was also invoked. The notion of "empowerment" was mentioned in two studies, although not defined. Additionally, two other studies also addressed related concepts. In Moala-Silatolu et al., CE "provides the necessary information for participation to their households as a rational decision and not based merely on persuasion" [35] and for Kisoka et al, it was used "to stimulate the endemic communities to own and sustain program interventions" [33].

Some researchers highlighted the lack of consensus on this term, regarding both its definition and purpose. Regarding its definition, Kaneto et al. pointed out that the difficulty lies in the ambiguity of the term participation: "There is the

difference between "participation" as used to denote the community simply accepting action that has been decided for them elsewhere, and the concept of participation as a process of community involvement in the planning, organization, operation, and control of health care as called for in the Declaration of Alma-Ata" [57]. Concerning purpose, different motivations drove researchers; Sahan et al. stated that "some researchers prioritize community engagement to promote the success of projects, in terms of study objectives or health outcomes, whereas others focus on its value in promoting ethical research practice" [53]. The same idea was formulated by Pell et al.: "some researchers, as in TME, prioritize its instrumental contribution to the success of studies, in terms of achieving specific research objectives or health outcomes; others focus on its intrinsic value for ethical research practice" [45].

It is noteworthy that seven studies did not explicitly delineate a clear objective in relation to the proposed CE intervention.

## Content of the CE interventions

Following the TIDieR checklist analysis [59], the level of detail in describing CE interventions varied between articles and interventions. No guidelines or checklists were used in articles to report on the development of interventions. Authors generally described the procedures and materials used in the intervention, locations, and providers. Conversely, articles rarely reported whether the intervention was modified during its course (describing, in that case, the changes implemented) or whether any strategies were used to improve compliance with the intervention or implementation fidelity.

**Content of the CE intervention.** In most cases (N = 23/24 interventions), content included information and education campaigns (IEC) to inform local authorities and target populations about the MDA intervention that is about to take place. IEC activities were organized at different scales: village-wide meetings, smaller group meetings, one-to-one meetings with village officials, informal discussions with individuals or with all household members, and village assemblies;... They were carried out in public places identified as important by the communities: health centers, schools, places of worship and markets.

Meetings with the administrative and health authorities (at different levels) were organized in most CE interventions (N = 19/24). They aimed to obtain their agreement and, less often, to involve them in the community engagement process. **Other informal authority figures or local opinion formers** were also targeted (N = 12/24): school administrators, police, teachers, shopkeepers, private sector health care providers, traditional healers and military staff. In most settings, religious leaders were approached. The research team included staff from the community in some cases.

**Communications media** (local radio, television spots, town criers), ads, posters, pamphlets, and flyers were frequently used to disseminate disease-related education (N = 17/24). In some cases, an event was organized in the village to raise awareness, such as music festivals and theatre in Cambodia or singing competition.

**Population targeted.** IEC initiatives usually target the whole village. More rarely, **specific groups** in the community were targeted: women, children at school to reach parents. Two studies reported outreach activities to target hard-to-reach population groups: mothers with young children, forest goers and migrants.

**Co-construction initiatives.** Some of the events did not aim solely at informing and educating but also left room **for co-construction initiatives** (N = 6/24). Some studies collaborated with populations at one or more stages of the intervention. They worked with community members to create materials for education and information activities [29,31]. Others relied on community members' decisions on how to conduct community involvement activities or how to organize drug distribution to individuals: the venue for distribution (at home or in a single location in the village) and the time of the day.

To do so, some interventions involved setting up a **steering committee** (N = 6/24). In Means et al., they presented results of formative research to key stakeholders during a 2-day meeting, who then created a group in charge of "defining implementation strategies and collaboratively describe each individual strategy's actor, action, action targets, dose, and temporality". In all the TME interventions, on each study site, a committee was formed of village leaders, village malaria

workers, and community volunteers [44–47,49–54,56]. They assisted the study team in designing and implementing community engagement. In Duamor et al., a Community Self-Monitoring (CSM) [38] selected drug distributors and decided how to motivate them, planned the drug distribution (when and how), and reported to the health service.

**Community health workers.**  The use of **community health workers** was particularly common (N = 19/24): recruited from the community, sometimes by members of the community**,** they were then trained and asked to help with drug distribution. They were either recruited specifically for the study or were already involved in the health system (to provide diagnosis and treatment of malaria for example). When they were already involved in local health care, they also assumed the role of an interface between the community and external parties, acting as an intermediary to guarantee the benefits of the intervention for the community. A few interventions required their help for awareness-raising campaigns: in the Cambodian setting of the TME, they were used to alert study team and leaders to rumors or perceived adverse reactions of MDA and help sensitize participants**.**

In terms of remuneration, there was a distinction between those who were paid or compensated and those who were not.

**Formative research.**  Some of the projects emphasized the importance of a detailed understanding of the local political and social structures in which the activities were to be carried out and **used formative research** to guide their implementation (N = 10/24). Researchers used qualitative methods: rapid ethnography and PAR activities (social mapping, transect walks and observations, key informant interviews, case interviews, informal focus group discussions, short surveys), meetings with local health authorities, villages leaders, community workers. Some authors of interventions of the TME project described an immersion of the research team in the community before and during MDA: "Study staff's participation in local social activities, such as funerals and local festivals, their presence in the villages and commensality–sitting round the same table and sharing traditional food–helped to build trust" [45].

**Timing.**  Regarding **timing**, most activities took place in the weeks preceding MDA (with little detail on the exact duration). A few initiatives continued their activities during the drug administration: Kings et al., continued distribution of pamphlets during MDA [28]. In the Cambodian TME intervention, the study team stayed in villages 7 days after each round of MDA to respond to complaints and concerns in small gatherings or house-to-house visits [44,46,47].

**Incentives. Financial incentives** were sometimes given to take part in activities or administering medication (N = 4/24). Some studies reported **non-monetary incentives** (N = 6/24), such as household gift packs and snacks. Finally, some teams adapted the incentives to the specific needs of the community: in the TME interventions, auxiliary care was provided for the duration of the study through a free clinic during each survey and round of MDA, and a piping system was set up to distribute drinking water from a nearby spring to village houses [44,46,47].

**Planned and unplanned variations.**  Some interventions were responsive (N = 5/24) and allowed adaptation according to events and feedback. Some research teams were particularly alert to concerns and rumors, especially about adverse effects, and focused their efforts on reactive information on potential side effects. Most frequently asked questions were identified so that educational messages could be adapted: with the study team answering listener call-in questions on radio shows, through door-to-door visits… Several interventions were also adapted according to the lessons learned after a year of intervention. Lessons learned were evaluated through a Knowledge, Attitudes, and Practices (KAP) questionnaire and distributors' interviews, through qualitative studies. Examples of adaptation included changes in the allocation of spending and efforts to reach hard-to-reach population. The Cambodian team of TME adapted the CE activities between two rounds of MDA seeing a decline in participation linked to trust issues: prior to round 3, the staff took less prominent roles in meetings and local health staff from the government-run facilities led the events [44,46,47].

**Malaria vs NTDs MDA interventions.**  A comparative analysis of CE strategies revealed differences between malaria and NTDs MDA interventions. Malaria-focused interventions with MDA were generally more recent, with 6 out of 8 published after 2013, compared to 4 out of 16 for NTDs. Co-construction initiatives were reported in 37% of malaria interventions versus 19% for NTDs. Hard-to-reach populations were specifically targeted in 25% of malaria interventions

but none of the NTDs. The use of formative research was also more common in malaria-related interventions (50% vs. 37%), as were both financial (25% vs. 12%) and non-financial incentives (37% vs. 19%). Responsiveness to community feedback was noted in 37% of malaria interventions, compared to 19% of those focused on NTDs.

## Methods used

According to the categories in the Mixed Methods Appraisal Tool [48], the evaluation methods used were descriptive quantitative (n = 8), mixed methods (n = 6), quantitative without randomization (n = 1) and qualitative (n = 17).

## Evaluation

Of the 24 CE interventions reported by the 32 studies, 6 were not evaluated.

**Design.** Few studies distinguished the role of CE from the overall assessment of all activities related to drug administration. Several authors pointed out that the evaluation was complicated by respondents' difficulties in disentangling community engagement from other study-related activities.

The findings most frequently evaluated were uptake rates, factors associated with uptake/affecting participation, lessons learnt and infection-related knowledge, attitudes, and practices. The factors studied for their influence on participation were the state of knowledge about the infection, the intervention, and the reasons for non-participation. Quantitative studies (N = 9) identified awareness of the drug distribution, participation in CE activities, knowledge about the infection, access to the drug and the perception that MDA was worthwhile as positively associated with drug intake. The findings from the population coverage surveys (treatment received) and compliance surveys (treatment ingested) yielded a wide range of results. Qualitative studies (N = 17) used semi-structured or in-depth interviews with leaders, community members or research team members. A study focused on a post-intervention evaluation of acceptability and adaptability, and another aimed to create a conceptual framework to acquire a deeper understanding of the success of CE activities. Six studies used both quantitative and qualitative methods to assess factors associated with participation.

**Findings.** The factors evaluated in the KAP surveys conducted after the implementation of CE interventions included the proportion and level of detail in the understanding of the key concepts related to the targeted infection (transmission, prevention, treatment), awareness of the drug distribution, sources of information, and the reason for non-participation. Krentel et al. conducted a before-after KAP study, noting improved knowledge regarding the infection and shifts in attitude (the idea that the infection is a health concern, attitude when feeling sick) [31].

**Qualitative data also reported factors associated with participation.** Factors often cited were the perception of the infection as a health concern, addressing concerns about adverse effects, managing them and rumors. Adapting to the local context and grasping concerns and attitudes through formative research (understanding history, past experiences, social dynamics and power relations) was crucial. Additionally, it appeared essential to spend sufficient time with community members, to facilitate the development of trust and the understanding of concerns. Furthermore, it was important to consider the importance of targeting various groups within the community, the necessity of adapting activities (season and work schedules, timing of distribution, and the importance of flexibility in implementation. Knowledge about the disease and awareness of the intervention was associated with taking the drug. Several studies emphasized the importance of relying on administrative and religious leaders and trusted members of the community and the fundamental role of community health workers. The role of financial incentives was often highlighted, yet their impact remained uncertain. In some studies, they were regarded as crucial, yet in Peto et al. their absence during two rounds of MDA did not result in a decline in participation [47]. Non-financial compensation, in the form of complimentary ancillary care, also played a role in TME interventions [44–47,49–54,56]. Non-financial incentives, such as access to free healthcare, can be understood as an acknowledgment that the intervention addresses broader health needs in the region. In an elimination context like the Greater Mekong Subregion (GMS), where malaria incidence is as low as 20 cases per 1,000 population per year, community health concerns increasingly extend beyond malaria itself.

**Challenges or limitations in implementation** frequently cited included reaching remote locations, top-down planned activities, lack of information, funding delays, human resource shortages, the difficulty to teach about concepts such as drug resistance, submicroscopic infections, asymptomatic disease and rumors.

Most of the evaluations focused on intermediate outcomes, such as the successful implementation of CE and the proportion of patients taking the medication. Only two studies assessed the intervention's effectiveness in reducing the infection rate, which is ultimately the primary goal of an MDA intervention [28,40]. And these studies did not attempt to distinguish the role of activities promoting community involvement from the rest of the MDA.

## Discussion

The analysis of these studies offered valuable insights into the difficulties inherent in assessing CE interventions. There were many different terms and definitions used for CE. The authors did not always define the notion, although it referred to a wide range of interventions. In our review, it was most often used to refer to activities related to health information or education. The objective was usually to increase the drug intake for the MDA intervention. When community participation was sought, it was for the implementation phase of the CE intervention, not for its design, development, or evaluation.

The issues surrounding CE are elucidated in this review. It is a multifaceted concept that is intricate to navigate. Primarily, we emphasize the necessity of defining the term. As identified by Brunton et al., a CE intervention can be situated within two distinct narratives: an 'utilitarian' or a 'social justice' approach [60].

The polysemy of the terms "participation" and "engagement" reflects their ambiguity. Participation can signify agreement to take the drug, but it can also signify participation in all the steps of the intervention (design, implementation, evaluation for example). In most studies, the term "community engagement" refers to participation in information and educational activities from a passive and instrumental perspective. The use of the term "community engagement" may warrant reflection when activities are limited to informing communities and obtaining their consent prior to an intervention. Some studies allowed participants flexibility in implementing the MDA intervention. Ultimately, when community involvement was genuinely present, the primary focus was on implementing the intervention. Notably, none of the studies referenced the community's involvement in the design or evaluation of the intervention. The authors did not perceive CE as a means of empowering communities through social and structural change, by enabling them to participate, negotiate, influence and hold accountable the institutions that affect them.

Despite the centrality of community engagement in MDA strategies, few studies assessed its direct impact on infection outcomes. The challenges of attributing causal effects to community engagement interventions on infection outcomes are classic in public health, as demonstrated in our study on the fight against dengue fever in Burkina Faso [61]. Like the analysis of the impact of empowerment interventions [62,63], community involvement interventions are necessarily complex [64]. It is, therefore, difficult, or even impossible, to isolate one dimension of an intervention to assess its impact on an infection outcome. However, the challenge is not only methodological but also epistemological in the context of MDA strategies, which are inherently complex. It is, therefore, more challenging to assess their impact using experimental or quasi-experimental studies. This type of intervention requires more holistic and systemic approaches to evaluate (and understand) its impact [65,66]. It would be interesting if malaria research teams interested in these evaluations could look at theory-based evaluation approaches (methods that use program theories to explain how and why interventions work) or realist epistemology (to understand what works, for whom, and under what circumstances) [67,68]. A rapid realist review of community engagement has been published, but it only concerned OECD countries [69]. There is undoubtedly an urgent need to apply this evaluation approach for LMICs, which still bear a heavy burden of malaria and yet have a long tradition of community engagement.

The differences observed between malaria and NTD interventions likely reflect a temporal shift. Most malaria studies have been conducted recently in research settings, where participatory and community engagement approaches are increasingly emphasised. Rather than a disease-specific contrast, this suggests an evolution in how CE is framed and implemented over time.

The lack of consensus on definitions of terms, the plurality of objectives and the complexity of the notion of CE render these studies complex to implement, evaluate and compare with the existing literature. It is possible to draw upon existing theories and conceptual frameworks to overcome these challenges. As an illustration, to distinguish between CE interventions and information or consent initiatives, we propose using Adhikari's definition of CE: "1) stakeholder and authority engagement; 2) enlisting local human resources; 3) utilizing formative research before conducting MDA trials; 4) responsiveness and adapting to local challenges as they arise; and 5) sharing control and leadership with the community in deciding and organizing activities" [50].

Secondly, mapping these activities onto a participation scale like Arnstein's ladder could help clarify the extent of community engagement. This scale is designed to categorize various participation models according to their participation level. In this model, true participation is only initiated once power is delegated or developed; other forms of participation are regarded as mere 'tokenism' and 'non-participation' [60].

A significant proportion of papers did not provide the information needed to complete the TIDieR checklist in terms of reporting the intervention, thereby limiting the possibility of assessing whether the results can be transferred from one setting to another. We propose to refer to validated criteria for describing complex interventions, such as the TIDieR adapted for Population Health and Policy interventions (TIDieR-PHP) [59].

The findings presented here prompt further reflection on the role of community engagement in MDA.

Beyond its instrumental value, community engagement in MDA interventions also serves a crucial ethical function. It contributes to individual autonomy by enabling informed decision-making and voluntary consent — a particularly sensitive issue in MDA, where individuals are often asymptomatic and may not perceive immediate personal benefit. Furthermore, the ethical justification of MDA at the population level relies on achieving high coverage to ensure collective protection and maximize the intervention's effectiveness. This is particularly important when administering drugs to largely uninfected individuals: the benefit/risk ratio becomes ethically acceptable only if sufficient population-level impact is achieved. As several authors have pointed out, including Parker & Allen (2013) and Singh et al. (2022), ethical acceptability of MDA hinges on transparency, community trust, and equitable participation. These considerations further support the need for meaningful, context-sensitive CE as a condition not only for operational success, but also for ethical legitimacy.

In the specific context of MDA to an asymptomatic population, CE could be the response to several challenges. The mass administration of drugs is marked by a colonial history [70], and the shadow of the recent vertical health policies of the COVID pandemic is still close at hand [71–73]. Any intervention should be designed from an ethical standpoint of social justice if they are to empower individuals to become active participants in the eradication of communicable diseases [74]. As sustainability is a key point to eradicating VBD, interventions should focus on community capacity building [75].

The effectiveness of such interventions relies on establishing trust between stakeholders, a process that is closely tied to a thorough understanding of the sociocultural context in which the intervention takes place.

We emphasize the importance of combining biomedical and social science expertise for a formative research approach that provides information on the local context, including social, political, and cultural factors [76]. Prior to the initiation and in situ research should be conducted to ascertain the social relationships within the community. A special effort needs to be made to identify the target groups, their health needs and the objectives of the intervention.

In the context of MDA interventions, where it is vital to reach all sections of the population, it is important to avoid the pitfall of considering a "community" as a homogeneous entity. This term often masks social fragmentation, and paying particular attention to hard-to-reach populations is essential. Some harm of CE can be social exclusion [60].

When possible, community members should be involved in the design of an intervention, not only take part in its delivery. The degree of collective decision-making should be reported in each part of the study [77,78].

Furthermore, the intervention should be responsive to events and feedback during the study period.

Despite its importance, the question of available resources and stable funding is rarely addressed in the literature, even though it is a central determinant of sustained community engagement [76].

The review identified the complexity of the issues and the scope for improvement in the implementation, evaluation, and reporting of such an intervention [79]. The method was further strengthened by the utilization of validated tools, including the PRISMA scoping review checklist, the TiDIER checklist and the MMAT checklist [59,80].

The review process presented several challenges. The analysis was further complicated by the fact that 11 articles described four distinct interventions. Moreover, the concept of community involvement has numerous synonyms. While we strived for comprehensive coverage, some relevant articles may have been missed, particularly as our search was restricted to French- and English-language publications and excluded grey literature.

## Conclusion

This scoping review highlights inconsistencies in implementing community engagement (CE) in mass drug administration (MDA) studies, with most interventions focusing on (passive) information dissemination rather than genuine (active) community involvement. The persistence of a public health approach that exploits community approaches is confirmed in the MDA field. Despite its ethical and social advantages, active participation of target populations in MDA interventions remains limited. Addressing barriers such as resource constraints, limited training, and lack of early community engagement is essential to enhance participatory approaches and strengthen the effectiveness and acceptability of MDA programs. The lack of clear definitions and standardized approaches limits the ability to assess and compare CE interventions. Adopting participatory frameworks that promote shared decision-making and deeper community involvement is essential to improve outcomes and reduce inequalities. Understanding local sociocultural contexts and ensuring the inclusion of marginalized groups are critical for enhancing participation and equity in MDA programs. Future interventions should prioritize sustainable, capacity-building approaches to maximize the effectiveness of CE in MDA efforts.

## Supporting information

**S1 Text. Protocol.**
(PDF)

**S2 Text. Search strategy.**
(PDF)

**S1 Data. Data extraction table.**
(XLSX)

**S2 Data. Excluded studies.**
(XLSX)

## Author contributions

**Conceptualization:** Anouk Chouaïd, Sarah Louart, El Hadj Ba, Jordi Landier, Valéry Ridde.

**Data curation:** Anouk Chouaïd, Sarah Louart, Valéry Ridde.

**Formal analysis:** Anouk Chouaïd, Valéry Ridde.

**Funding acquisition:** Adama Faye, Jordi Landier, Valéry Ridde.

**Investigation:** Anouk Chouaïd.

**Methodology:** Anouk Chouaïd, Valéry Ridde.

**Project administration:** Adama Faye, Jordi Landier, Valéry Ridde.

**Supervision:** Adama Faye, El Hadj Ba, Jordi Landier, Valéry Ridde.

**Validation:** Sarah Louart, Jordi Landier, Valéry Ridde.

**Visualization:** Anouk Chouaïd, Jordi Landier, Valéry Ridde.

**Writing – original draft:** Anouk Chouaïd.

**Writing – review & editing:** Anouk Chouaïd, Sarah Louart, Adama Faye, El Hadj Ba, Jordi Landier, Valéry Ridde.

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
