## [Decision Letter · Decision Letter 0]

12 May 2025

Community engagement in mass drug administration participatory interventions: a scoping review

Dear Dr. Chouaïd,

Thank you for submitting your manuscript to PLOS Neglected Tropical Diseases. After careful consideration, we feel that it has merit but does not fully meet PLOS Neglected Tropical Diseases's publication criteria as it currently stands. Therefore, we invite you to submit a revised version of the manuscript that addresses the points raised during the review process.

Please submit your revised manuscript within 60 days Jul 11 2025 11:59PM. If you will need more time than this to complete your revisions, please reply to this message or contact the journal office at plosntds@plos.org. Please include the following items when submitting your revised manuscript:

We look forward to receiving your revised manuscript.

Kind regards,

Olaf Horstick, FFPH(UK)

Academic Editor

Justin Remais

Section Editor

Shaden Kamhawi

co-Editor-in-Chief

Paul Brindley

co-Editor-in-Chief

**Journal Requirements:**

At this stage, the following Authors/Authors require contributions: Anouk Chouaïd. Please ensure that the full contributions of each author are acknowledged in the "Add/Edit/Remove Authors" section of our submission form.

- ® on page: 8.

4) Please ensure that all Figure files have corresponding citations and legends within the manuscript. Currently, Figure 4 in your submission file inventory does not have an in-text citation. If the figure is no longer to be included as part of the submission, please remove it from the file inventory.

5) We have noticed that you have uploaded Supporting Information files, but you have not included a list of legends. Please add a full list of legends for your Supporting Information files after the references list.

6) In the online submission form, you indicated that "The data that support the findings of this study are available from the corresponding author, AC, upon request." All PLOS journals now require all data underlying the findings described in their manuscript to be freely available to other researchers, either

1. In a public repository

2. Within the manuscript itself

3. Uploaded as supplementary information.

**Comments to the Authors:**

**Please note that one of the reviews is uploaded as an attachment.**

**Reviewers' Comments:**

Reviewer's Responses to Questions

**Key Review Criteria Required for Acceptance?**

**Methods**

-Are the objectives of the study clearly articulated with a clear testable hypothesis stated?

-Is the study design appropriate to address the stated objectives?

-Is the population clearly described and appropriate for the hypothesis being tested?

-Is the sample size sufficient to ensure adequate power to address the hypothesis being tested?

-Were correct statistical analysis used to support conclusions?

-Are there concerns about ethical or regulatory requirements being met?

Reviewer #2: Authors reported their study according to PRISMA guidelines and clearly detailed the selection process and exclusion criteria (figure 1). Author contribution is explicitly acknowledged throughout most of the methods section. Study design is justified and seems appropriate to address stated objectives.

Significant aspects are missing from the methods section:

- Definitions of CE and MDA used for the review remain unclear, compromising the understanding of the search strategy, inclusion of publications and data analysis - e.g., "We adopted a broad definition of CE" => please provide the definition and synonyms used

- Please provide a representative search string or synonyms used in your search strategy, for both semantic clusters (CE and MDA): this is essential for reproducibility of the study and should be provided in the manuscript, not only in the pre-registration

- Methods for data analysis are not described sufficiently in the manuscript, and the reference to the supplementary material is missing - e.g., "For data extraction and analysis, items from TIDIeR and MMAT checklist theoretical framework were used" => please explain which elements and the rationale for the choice of these elements, how you proceeded with their evaluation, and who conducted this work

Reviewer #3: objectives and methods reasonably met (see below for additional comments on the scope).

**Results**

-Does the analysis presented match the analysis plan?

-Are the results clearly and completely presented?

-Are the figures (Tables, Images) of sufficient quality for clarity?

Reviewer #2: Overall, the results are well structured and match the stated objectives of the study.

Points to improve include:

- Unfortunately, clear differences between MDA for NTDs and MDA for malaria are not acknowledged nor reflected in the results, which weakens the output of this publication.

- "Out of 24 interventions, 16 focused on NTDs" and among 8 malaria interventions, 4 belonged to the same research project: this is not reflected in the manuscript, which heavily focuses on malaria, including in the examples given in the results section.

- Meaning of reported numbers are unclear: do they refer to interventions or publications? E.g., "In most cases (N=30)". This is also valid for the following paragraphs and would be important to clarify.

- Refer to supplementary materials for details on data analysis in the main text

Reviewer #3: Yes, but with comments.

**Conclusions**

-Are the conclusions supported by the data presented?

-Are the limitations of analysis clearly described?

-Do the authors discuss how these data can be helpful to advance our understanding of the topic under study?

-Is public health relevance addressed?

Reviewer #1: conclusions and discussion require changes.

the comment enclosed.

Reviewer #2: The discussion section is easy to read, clear and well-structured. Relevance of this study for public health is clearly stated and a list of recommendations to improve CE in MDA is provided and very helpful to improve further interventions (from "we emphasize the importance of combining" to "stability of funding, which is a key determinant of community participation"). You could emphasize further that this is a big output of your study. Limitations are stated, but should also include the focus on two languages only (French and English).

Suggestion of further discussion points to improve the manuscript:

- CE is framed as a way of reaching marginalised groups/all sections of the population - but specific groups were systematically excluded from the analysed interventions ("pregnant women, sick people, children under age of two"), this would benefit from being discussed

- In the results section, authors state that most studies do not evaluate the effect of CE on reduction of infection, which is the ultimate goal of MDA. The discussion would benefit from a greater analysis of this point: why is it so, what are the difficulties, what are the implications of not knowing the effect of CE?

- Ethical aspects of CE could be addressed more: including the importance of CE (and information campaigns mentioned) for autonomy and informed consent in MDA; the importance of large coverage to reach reduction of infection and importance of this to make intervention ethically sound, balancing out the benefit/risk ratio for the uninfected majority. Refer to literature on MDA ethics.

- Please discuss the framing of "free clinics" and "providing drinking water" as incentives to participation as these are basic human needs (and essential to the roll-out of the intervention)

Reviewer #3: Yes, but with caveats mentioned below.

**Editorial and Data Presentation Modifications?**

Reviewer #2: - Figure 1 lacks arrows to guide reader through the selection pathway (might be an issue of formatting), also consider including "title-abstract screening" and "full-text screening" to make distinction between steps explicit

- Figure 2 would be more readable and accurate as a bar chart

- Figure 4 is not referred to in the main text

- Figures 3 and 4 could be merged which would increase the strength of the figure

- Figure 5 needs reviewing, the given information is not understandable at first glace: this information can either be provided in the text, or the figure should be transformed into a histogram or similar with readable percentages

- A lot of abbreviations are introduced that are not used much and could be avoided to improve readability (e.g., DHA-PPQ, GMS etc.)

**Summary and General Comments**

Reviewer #2: This study is a scoping review about the use and contribution of community engagement in Mass Drug Administration (MDA) interventions. Authors focus on MDA conducted in the context of malaria and NTDs. Community engagement is a pivotal aspect to consider in MDA, both due to the ethical implications of delivering medication to people without prior diagnosis and due to the necessity of a large population coverage to reap the benefits of the measure. Moreover, the authors highlight a recent surge of publications on the topic of community engagement in MDA, which demands updated knowledge on these interventions. This research is highly needed to inform the design of MDA interventions. Authors present interesting results and a well-structured discussion that clearly articulates the relevance of results for Public Health and provides tangible recommendations.

However, in its current form, the study presents significant shortcomings. Firstly, the rationale for the scope of the study is unclear. The authors include MDA interventions for NTDs and malaria, without explaining the rationale for analysing them jointly. Goals of MDA and methods used differ according to the disease (please refer to the WHO NTD roadmap 2021-2030, which outlines NTDs amenable to MDA, including levels of targeted control - e.g., control/ eliminate/eradicate, drugs of choice and target populations). Moreover, for many NTDs, MDA is a well-established public health measure, that has been carried out for decades in at-risk communities, whereas MDA for malaria seems to be carried out more frequently in research studies, is less common and not the primary strategy used to tackle the disease in at-risk settings. This pivotal difference is not acknowledged nor discussed in the manuscript. Additionally, most examples used throughout the manuscript focus on MDA for malaria, with MDA for NTDs being seldom addressed (which is questionable, both scientifically and given the scope of this journal) Clarifying the scope and justification for the selection of diseases and interventions is essential to improve the quality and output of this study. Authors could consider the following options:

- Clarify why malaria and NTDs are grouped in the analysis, as MDA has different goals for these diseases and implementation contexts: why can these interventions be compared? Why are other MDAs excluded (e.g., Mass Drug Administration of Azithromycin to Children under Five Years of Age to Promote Child Survival)?

- Separate the analysis of malaria MDA and NTD MDA and compare the results

- Focus the analysis on malaria and discuss results against examples of NTD MDA

Other areas for general improvement include:

- Introduction structure and content: the introduction lacks a clear structure and flow, making it difficult to follow. Authors should reconsider and restructure the content of the introduction, to enable good understanding of the background and scope of this research. This links to the previous comment about the scoping of this review, which should be clearly explained in the introduction.

- Definitions: general definitions on which the research is based are missing. Both key concepts of the study, MDA and Community Engagement, are not clearly defined by the authors. Clarity on the definition used by the authors is essential to understand the study scope and methods (including the search strategy, making it reproducible) - this should be clarified early on in the introduction.

-A lot of sentences would benefit from being more specific, and key take-aways from the references should be made explicit - e.g., "Community-based participatory (CBP) intervention theory suggests that engaging community members as collaborators in the interventions to reduce health disparities is powerful on multiple levels" => please explain what levels

-Methods for data analysis should be detailed more, to enhance understanding and make the study reproducible

-Authors could consider discussing the ethical relevance of Community engagement in MDA in greater detail

Reviewer #3: Thank you for the opportunity to review the paper Community engagement in mass drug administration participatory interventions: a scoping review. The authors have evaluated how community engagement is done looking at this from the perspective of whether communities were “actively engaged”. MDA is an important tool for many diseases and coverage and equity are important factors, so this is an interesting topic and the last review was done nearly a decade ago.

Major comments

The first key finding mentioned is: “Adopting participatory frameworks that promote shared decision-making and deeper community involvement is essential to improve outcomes and reduce inequalities.” Where is the evidence from the scoping review to support this statement? Or, is this based on other sources or opinions, in which case it is not really a finding from the review itself.

Have the authors used an appropriate strategy for identifying the literature? They find list only 24 interventions across 20 countries, and most are recent. A quick search on PubMed using the terms “Mass drug administration” & participation, returns 695 results. Many of which do not appear in the scoping review but appear to be relevant. E.g. 10.1179/000349804X3135 10.1179/136485906X105598 . The term community engagement was not very common before the year 2000 https://pubmed.ncbi.nlm.nih.gov/?term=%22community+engagement%22&sort=date . Moreover, all the papers that were reviewed in detail are from the published literature. But mass drug administrations have been used for disease control and elimination for many decades. For example, Ivermectin has been used billions of times as part of MDA campaigns. There must be a huge grey literature on this?

The article concludes: “The findings emphasize the importance of context-specific approaches, especially in addressing local sociocultural dynamics and including marginalized populations. Future interventions should prioritize sustainable capacity-building and adopt participatory frameworks that promote shared decision-making. Addressing these challenges could enhance the effectiveness of MDA campaigns and improve health outcomes in affected communities.” This is an easy statement to agree with, and the authors aren’t the first people to have noticed this. But if tailored, participatory engagement is not widely done or reported, why might this be? Is it always practicable in trial contexts, is it cost-effective in national programmes? Has it been tried but it actually doesn’t make as much difference in practice as it does in theory? What are the key steps? Are MDAs being done by teams without cultural knowledge of local communities, or often by people with excellent knowledge and experience of working with the communities they are seeking to persuade to participate in the MDA. Are there examples of participatory CE where it hasn’t worked or is it an unalloyed good?

“Future interventions should prioritize sustainable, capacity-building approaches to maximize the effectiveness of CE in MDA efforts.” I agree, and expect most people who have been involved in conducting MDA would too. This scoping review appears to have been performed well and the findings are worth publishing, but the tone of the article is somewhat preachy and virtue-signalling, and this detracts from its impact. The point about participatory community engagement has been before, so I suggest the article would benefit from being toned down a bit.

The Author Summary begins “The persistence of a public health approach that exploits community approaches is confirmed in the MDA field.” Please rephrase this, it is very hard to follow. Actually, it would be worth having a native English speaker proof read the whole article, as many of the sentence constructions are odd.

Apologies if these comments seem overly peevish. Again, this is a worthwhile review, properly done, with findings that will be of interest to a wide audience.

PLOS authors have the option to publish the peer review history of their article (what does this mean? ). If published, this will include your full peer review and any attached files.

**Do you want your identity to be public for this peer review?** For information about this choice, including consent withdrawal, please see our Privacy Policy .

Reviewer #1: No

Reviewer #2: No

Reviewer #3: No

**Figure resubmission:**

**Reproducibility:**



---

## [Decision Letter · Decision Letter 1]

11 Aug 2025

Response to Reviewers
Revised Manuscript with Track Changes
Manuscript

Shaden Kamhawi

co-Editor-in-Chief

Paul Brindley

co-Editor-in-Chief

**Journal Requirements:**

1) We have noticed that you have uploaded Supporting Information files, but you have not included a list of legends. Please add a full list of legends for your Supporting Information files after the references list.

2) We note that your Data Availability Statement is currently as follows: "All data underlying the findings described in the manuscript is available either within the manuscript itself or in the documents provided as supplementary information.". Please confirm at this time whether or not your submission contains all raw data required to replicate the results of your study. Authors must share the “minimal data set” for their submission. PLOS defines the minimal data set to consist of the data required to replicate all study findings reported in the article, as well as related metadata and methods (https://journals.plos.org/plosone/s/data-availability#loc-minimal-data-set-definition).

3) As required by our policy on Data Availability, please ensure your manuscript or supplementary information includes the following:

4) Kindly revise your competing statement to align with the journal's style guidelines: 'The authors declare that there are no competing interests.'

**Reviewers' comments:**

**Key Review Criteria Required for Acceptance?**

**Methods**

-Are the objectives of the study clearly articulated with a clear testable hypothesis stated?

-Is the study design appropriate to address the stated objectives?

-Is the population clearly described and appropriate for the hypothesis being tested?

-Is the sample size sufficient to ensure adequate power to address the hypothesis being tested?

-Were correct statistical analysis used to support conclusions?

-Are there concerns about ethical or regulatory requirements being met?

Reviewer #1: The objectives are clear

The study design is appropriate

Sample size is adequate.

Statistical test is fine.

No ethical consideration

Reviewer #2: (No Response)

**Results**

-Does the analysis presented match the analysis plan?

-Are the results clearly and completely presented?

-Are the figures (Tables, Images) of sufficient quality for clarity?

Reviewer #1: Analsis is appropriate.

Results are clear

Reviewer #2: (No Response)

**Conclusions**

-Are the conclusions supported by the data presented?

-Are the limitations of analysis clearly described?

-Do the authors discuss how these data can be helpful to advance our understanding of the topic under study?

-Is public health relevance addressed?

Reviewer #1: Conclusions are supported by data.

Limitations are stated.

The public healthrelevance is addressed.

Reviewer #2: (No Response)

**Editorial and Data Presentation Modifications?**

Reviewer #1: Accept

Reviewer #2: - Figure 1: in the box "References removed", heading "other reason", the number is missing

- Results, under the section "Qualitative data also reported factors associated with participation", one bracket is missing and one sentence is double (same meaning twice): "Knowledge was often identified as a significant factor in determining whether an individual would take the drug. Knowledge about the disease and awareness of the intervention was associated with taking the drug."

**Summary and General Comments**

Reviewer #1: The question raised in intial review has been addressed

Reviewer #2: The manuscript has significantly improved, especially the introduction and the discussion. The authors have well addressed the reviewers' comments.

A few minor suggestions for the authors' consideration:

- The article has a clear focus on malaria MDAs, as most publications selected were from this field and the examples taken in the results section are mostly malaria MDAs. Authors hint that this could be due to CE being most described in recent publications. The authors could clearly state and explain this focus on malaria in the introduction for more transparency with the reader.

- It seems that CE is a relatively new practice or concept described in publications on MDA. Why is it the case? It would be interesting to see this discussed.

- The authors are strongly favorable to active participation of MDA target populations in intervention design and evaluation (rather than being passive information recipients). This seems to be ethically and socially sound, but seems to not have been done frequently. Why is it the case? What could be barriers to implementing active participation in MDA interventions and how could these be overcome?

- In the discussion, authors state that “It would be interesting if malaria research teams interested in these evaluations could look at theory-based evaluation approaches or realist epistemology”. Please briefly explain the terms "theory-based evaluation approaches" and "realist epistemology" as these are not widely known.

PLOS authors have the option to publish the peer review history of their article (what does this mean? ). If published, this will include your full peer review and any attached files.

**Do you want your identity to be public for this peer review?** For information about this choice, including consent withdrawal, please see our Privacy Policy .

Reviewer #1: **Yes: ** Kaliaperimal Karthikeyan

Reviewer #2: No

**Figure resubmission:****Reproducibility:** To enhance the reproducibility of your results, we recommend that authors of applicable studies deposit laboratory protocols in protocols.io, where a protocol can be assigned its own identifier (DOI) such that it can be cited independently in the future. Additionally, PLOS ONE offers an option to publish peer-reviewed clinical study protocols. Read more information on sharing protocols at https://plos.org/protocols?utm_medium=editorial-email&utm_source=authorletters&utm_campaign=protocols

---

## [Decision Letter · Decision Letter 2]

10 Nov 2025

Dear MD Chouaïd,

We are pleased to inform you that your manuscript 'Community engagement in mass drug administration participatory interventions: a scoping review' has been provisionally accepted for publication in PLOS Neglected Tropical Diseases.

Best regards,

Olaf Horstick, FFPH(UK)

Academic Editor

Sitara Ajjampur

Section Editor

Shaden Kamhawi

co-Editor-in-Chief

Paul Brindley

co-Editor-in-Chief

Reviewer's Responses to Questions

**Key Review Criteria Required for Acceptance?**

**Methods**

-Are the objectives of the study clearly articulated with a clear testable hypothesis stated?

-Is the study design appropriate to address the stated objectives?

-Is the population clearly described and appropriate for the hypothesis being tested?

-Is the sample size sufficient to ensure adequate power to address the hypothesis being tested?

-Were correct statistical analysis used to support conclusions?

-Are there concerns about ethical or regulatory requirements being met?

Reviewer #2: Accept

**Results**

-Does the analysis presented match the analysis plan?

-Are the results clearly and completely presented?

-Are the figures (Tables, Images) of sufficient quality for clarity?

Reviewer #2: Accept

**Conclusions**

-Are the conclusions supported by the data presented?

-Are the limitations of analysis clearly described?

-Do the authors discuss how these data can be helpful to advance our understanding of the topic under study?

-Is public health relevance addressed?

Reviewer #2: Accept

**Editorial and Data Presentation Modifications?**

Reviewer #2: Regarding our previous comment: "The article has a clear focus on malaria MDAs, as most publications selected were from this field and the examples taken in the results section are mostly malaria MDAs. Authors hint that this could be due to CE being most described in recent publications. The authors could clearly state and explain this focus on malaria in the introduction for more transparency with the reader.", we think that the authors misunderstood the point. Malaria MDA is overrepresented in this study, as compared with NTD MDA (the latter being clearly more frequently described in the literature, as it has been conducted for a longer time already). This is fine, as long as the authors:

(i) clearly state that this is the case

(ii) explain why this is the case. Our hypothesis is that CE is more present in malaria MDA literature (as compared to NTD MDA literature) because both malaria MDA and CE for MDA are relatively newly described. Older publications on NTD MDA might not describe CE because it was not relevant at that time... and maybe implementation lags behind compared to newly designed interventions, as are malaria MDAs.

**Summary and General Comments**

Reviewer #2: Accept

PLOS authors have the option to publish the peer review history of their article (what does this mean? ). If published, this will include your full peer review and any attached files.

**Do you want your identity to be public for this peer review?** For information about this choice, including consent withdrawal, please see our Privacy Policy .

Reviewer #2: No

---

## [Editor Report · Acceptance letter]

Dear MD Chouaïd,

We are delighted to inform you that your manuscript, "Community engagement in mass drug administration participatory interventions: a scoping review," has been formally accepted for publication in PLOS Neglected Tropical Diseases.

Best regards,

Shaden Kamhawi

co-Editor-in-Chief

Paul Brindley

co-Editor-in-Chief
